# Global land use changes are four times greater than previously estimated

Karina Winkler 1,2✉, Richard Fuchs 2, Mark Rounsevell2,3,4 & Martin Herold 1

Quantifying the dynamics of land use change is critical in tackling global societal challenges such as food security, climate change, and biodiversity loss. Here we analyse the dynamics of global land use change at an unprecedented spatial resolution by combining multiple open data streams (remote sensing, reconstructions and statistics) to create the HIstoric Land Dynamics Assessment + (HILDA +). We estimate that land use change has affected almost a third (32%) of the global land area in just six decades (1960-2019) and, thus, is around four times greater in extent than previously estimated from long-term land change assessments. We also identify geographically diverging land use change processes, with afforestation and cropland abandonment in the Global North and deforestation and agricultural expansion in the South. Here, we show that observed phases of accelerating (~1960–2005) and decelerating (2006–2019) land use change can be explained by the effects of global trade on agricultural production.

[1] Laboratory of Geoinformation and Remote Sensing, Wageningen University & Research (WUR), Wageningen, The Netherlands. [2] Land Use Change & Climate Research Group, IMK-IFU, Karlsruhe Institute of Technology (KIT), Karlsruhe, Germany. [3] Institute of Geography & Geo-ecology (IFGG), Karlsruhe Institute of Technology (KIT), Karlsruhe, Germany. [4] School of GeoSciences, University of Edinburgh, Edinburgh, UK. ✉email: karina.winkler@kit.edu

About three-quarters of the Earth's land surface has been altered by humans within the last millennium[1,2]. Successfully tackling global sustainability challenges such as climate change, biodiversity loss and food security depends on land use change, since it strongly affects carbon sources[3] and sinks[4,5], causes habitat loss[6] and underpins food production[7]. In particular, the mitigation potential of land use activities, including those related to forests and agriculture, has been recognised as essential in meeting climate targets under the Paris Agreement, making land use a central component of many international policy debates[2,8]. Therefore, quantifying and understanding global land use change and its spatiotemporal dynamics is critical in supporting these debates.

Yet, in spite of its societal relevance, understanding how global land use/cover (LUC) has changed across space and through time is limited by a lack of comprehensive data and the large uncertainties within existing LUC reconstructions[9,10].

Even in the age of satellites, 'big data' and a growing trend of opening access to information, LUC data are still constrained by fragmented content, varying scales, a lack of spatial or temporal detail and inconsistent time series[11,12]. Satellite remote sensing refers to land cover (the biophysical properties of a land surface, e.g. grassland) and provides high spatial resolution, but short temporal coverage. In contrast, inventories and statistics mostly concern land use (the purpose for, and activities by which humans utilise land, e.g. grazing, cropping), encompass long time spans, but are bound to administrative units and, thus, lack spatial detail. Each data source on its own lacks one critical component—space, time or theme—and, thus, is unable to capture the full scale of land use dynamics.

Existing global, long-term land use reconstructions often rely on only a few observational data streams and are built on assumptions concerning, for example, the allocation of cropland (HYDE3.2[13], LUH2[14]) or wood harvests (LUH2[14]). They also have rather coarse spatial resolutions of up to 0.25 degree (LUH2[14]) and limited land use categories (SAGE cropland[15], HYDE3.2[13]). Although recent progress was made by GLASS-GLC[16] in assessing long-term, land cover change at an unprecedented spatial resolution (5 km) and temporal coverage (1982–2015), GLASS-GLC only refers to land cover (not land use) and relies on a single satellite sensor (AVHRR) as a data source. More importantly, none of the existing data on land use change fully account for gross change, in other words, all of the land transitions between LUC categories that occur during a given time period. However, identifying gross changes in land use dynamics is essential when quantifying the climatic and environmental impact of LUC change[9].

To analyse and better understand the spatiotemporal dynamics of global land use change, we combined multiple, high-resolution remote sensing data (see Supplementary Table 1) with long-term statistical data streams (FAO land use[17] and population[18]) to assess annual changes in LUC from 1960 to 2019 at a spatial resolution of 1 km. Based on open datasets, we developed a model called HILDA + (Historic Land Dynamics Assessment + , https://landchangestories.org/hildaplus-mapviewer/), which harmonises spatially explicit LUC information with land use inventories at the national scale and allocates these changes to the global land surface. The approach fully incorporates data-derived, annual gross changes between six LUC categories: urban, cropland, pasture/rangeland, forest, unmanaged grass/shrubland, sparse/no vegetation (see Supplementary Table 2). This enables the quantification of the spatial extent of land use change in unprecedented detail and provides tracking of the annual dynamics through time.

In this paper, we present the gains and losses in major LUC categories, identify different land use change patterns and compare these across the globe.

## Results and discussion

**Spatial extent and diverging patterns of global land use change.** We estimate that 17% of the Earth's land surface has changed at least once between the six land categories from 1960 to 2019 (see Fig. 1). When summing all of the individual change events (including areas of multiple change), the total land change extent is 43 million km², which is almost a third of the global land surface. This means that, on average, a land area of about twice the size of Germany (720,000 km²) has changed every year since 1960.

We identify a global net loss of forest area of 0.8 million km², but an expansion in global agriculture (i.e. cropland and pasture/rangeland) of 1.0 and 0.9 million km², respectively. However, the

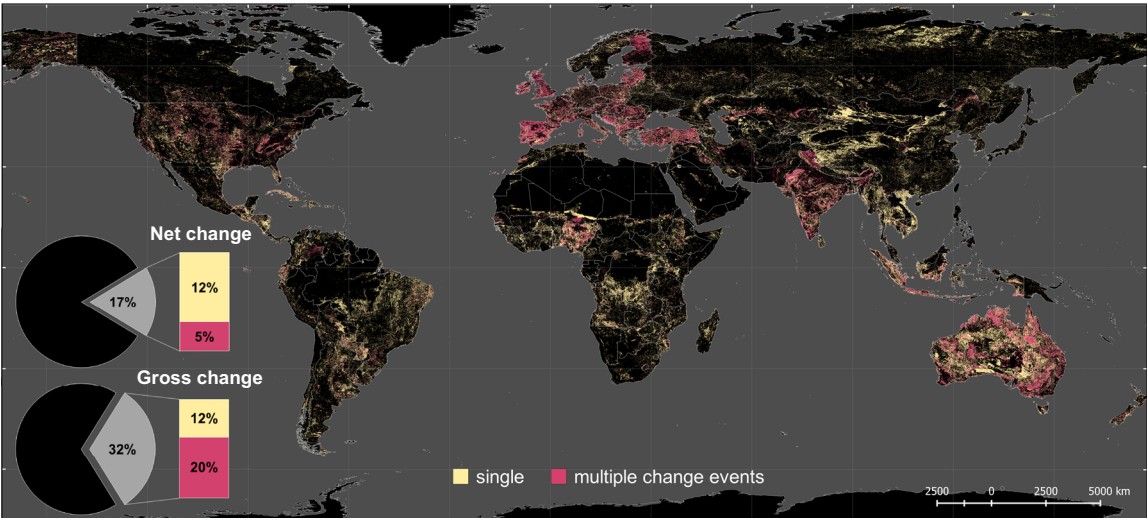

**Fig. 1 Spatial extent of global land use/cover change.** Share of the total land surface without (net change) and with consideration of multiple changes (gross change) between six major land use/cover categories (urban area, cropland, pasture/rangeland, forest, unmanaged grass/shrubland, non-/sparsely vegetated land) in 1960–2019. The spatial extent of land use/cover change is displayed in yellow (areas with single change events) and red (areas with multiple change events).

global trends in land use change conceal many regionally different trajectories. Whereas forest areas in the Global North (including China) have increased, forest areas in developing countries of the Global South have strongly decreased. The North-South difference in gains and losses of forests, is the opposite for global cropland areas, which have decreased in the Global North and increased in the Global South. The difference between North and South is less pronounced for pasture/rangeland change, since pasture expansion both in China and Brazil accounts for a major part of the global land area (see Fig. 2). These globally diverging land use change processes are supported by numerous studies, e.g. forest gain caused by political reforestation incentives in China[19–21], agricultural land abandonment in Europe[22] and the United States[23–25], climate-induced vegetation shifts in Siberia[26–28] and woody encroachment of rangelands in the United States [29]and Australia[30]. Conversely, tropical deforestation has occurred for the production of beef, sugar cane and soybean in the Brazilian Amazon[31,32], oil palm in Southeast Asia[33–36] and cocoa in Nigeria and Cameroon[37–39]. Furthermore, rangelands have expanded widely into marginal lands in China[19,40].

By separating land use change into areas with a single change (e.g. deforestation) or multiple change events (e.g. crop-grass rotation), we see clear patterns across the globe (see Fig. 1). Of all land transitions, 38% are single change events, which are most evident in developing countries of the Global South. Around half of the areas with single change events (48%) comprise agricultural expansion, which can be seen, for example, in the expanding pastureland of China or in tropical deforestation in the Amazon. Multiple change events make up 62% of all land transitions. In contrast to single changes, multiple changes dominate in the developed countries of the Global North (e.g. in Europe, the United States, Australia) and rapidly growing economies (e.g. Nigeria, India). Here, agricultural intensification, as in the EU and the United States and/or major transitions in the agricultural sector, for example, the switch from subsistence to commodity crops in Nigeria[41], have taken place over the last decades. Of all multiple change events, 86% are agricultural land use changes (land transitions related to cropland or pasture/rangeland). Some of these changes are directly or indirectly linked to land management and agricultural intensification. Cropland-pasture/rangeland transitions (11% of all multiple change events) can indicate areas of crop rotation or mixed crop-livestock systems as in the United States, Australia and in Europe[42,43]. Most multiple changes (75%) take place between managed and unmanaged land such as the abandonment of cropland, e.g. due to agricultural intensification on more suitable land as in Post-Soviet Eastern Europe[44], rangeland-shrub encroachments as in rotational grazing systems in Australia[45] or the Mediterranean as well as transitions between agricultural land and forest as in agroforestry systems in western Europe[46].

**Temporal dynamics of global land use change and its relation to globalised markets**. The rate of global land use change was not constant over time. In analysing the temporal dynamics, we identify two different phases: (1) an acceleration phase with an increasing rate of change from 1960 to 2004; and (2) a decreasing rate of change from 2005 to 2019 (see Fig. 3). The transition from constant to rising rates of land use change has been discussed in the context of shifting global food regimes and coincides with a period when global food production changed from agro-technological intensification (driven by the Green Revolution in the 1960s) to the production for globalised markets and increasing trade, especially during the 1990s[47,48]. We find this acceleration phase to be more distinct in regions of the Global

South, as observed in South America, Africa and Southeast Asia (see Fig. 3), where production and export of commodity crops have increased, most strikingly since the 2000s (see Supplementary Figs. 1 and 2). The growing influence of tele-connected markets is found to be a major driver of land use change, particularly deforestation for commodity crops in the Global South[39]. This offshoring of land use change from the Global North to the South is evident in the growing proportion of cropland in the countries of the Global South used for export and consumption outside of their territories[49].

However, the data suggest a rather abrupt change to decreasing rates of land use change in the period from 2005, which is most evident in Africa and South America (see Fig. 3), regions of the Subtropics and Tropics (see Supplementary Fig. 3). We hypothesise that the transition from accelerating to decelerating land use change is related to market developments in the context of the global economic and food crisis 2007–2009. Before the crisis, rising demand for food, animal feed and biofuels as well as increasing oil prices (reaching an all-time high in 2008 at $145.31 per barrel of Crude[50]) stimulated global agricultural production, which enhanced global land use change[51]. In particular, high oil prices made bioenergy crops more competitive and profitable compared to fossil fuels. Increasing demand, mostly in the developed countries of the Global North, spurred bioenergy crop expansion in the Global South (e.g. production of oil crops in Ghana, Argentina, Brazil and Indonesia, see Supplementary Fig. 1). Biofuel policies, climatic extremes and export bans led to global food price spikes in 2007–2008[52] and in 2010[53,54], which raised concerns about food security in many import-dependent countries and rapidly growing economies (e.g. the EU, China or India). A wave of large-scale, transboundary land acquisitions and foreign investments in agriculture emerged, mostly targeting sub-Saharan Africa, Southeast Asia and South America[48,55,56]. This development is reflected in the sudden increase in the rate of land use change (during 2000–2005), ensuing fluctuations (during 2006–2010) and sharp decrease (after 2010) in countries of the Global South, e.g. Brazil, Argentina or Ethiopia (see Supplementary Fig. 4). We find that the observed slowdown of global land use change after the economic crisis 2007–2009 is mainly caused by a decline in agricultural expansion in the countries of the Global South, particularly pronounced in Argentina, Ghana and Ethiopia (see Supplementary Fig. 5). We postulate that the global deceleration of land use change is related to market mechanisms during the economic crisis. With the economic boom coming to an end during the Great Recession, the global demand for commodities dropped. Countries which focussed on the production of commodity crops for global markets prior to the crisis (e.g. Argentina, Brazil, Ghana or Indonesia), no longer found buyers for their goods, reduced agricultural production and, thus, the rate of agricultural land expansion. The observed sharp decline in the rate of land use change, especially in Africa (see Fig. 3), may be further caused by a decrease in the number and size of global land acquisitions after the financial crisis in 2007–2009. Since then, hedge funds in land became less common[57] and concerns were raised about unsustainable practices related to transboundary land acquisitions (e.g. land/water degradation and displacement of rural labour)[52,57]. Resulting incentives from international organisations and exporting countries to limit land trade may have led to the recent decline in large-scale land acquisitions[57].

Aside from globalised trade, other important drivers of land change dynamics, which have increasingly influenced the rate of land use change during the deceleration phase, are climate change and its associated impacts such as extreme events, drought and floods. Agricultural land use has been affected by droughts in West[58] and Eastern Africa[59] during the 2000s, which can be observed in the strong decline in the rate of land use change in

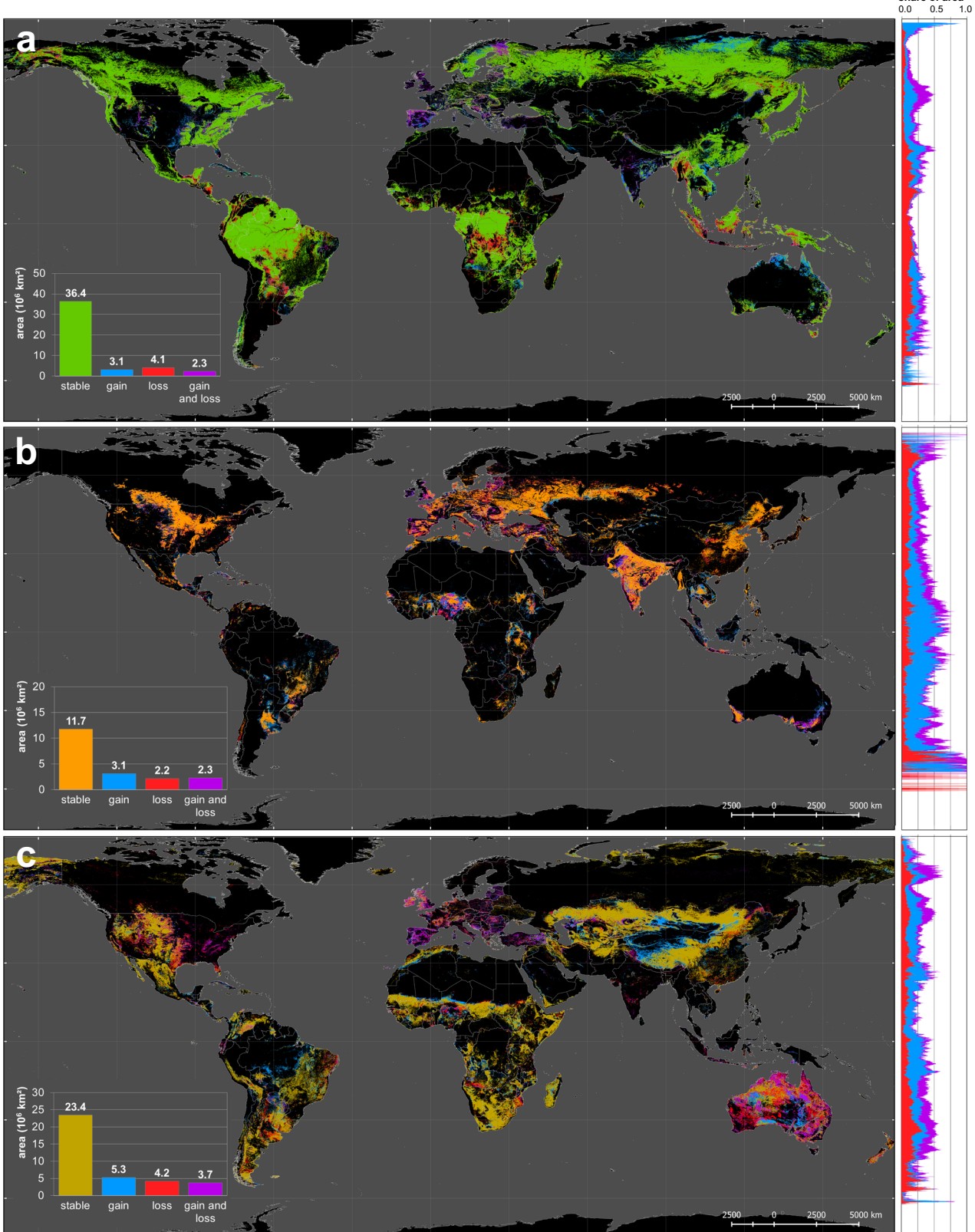

**Fig. 2 Global forest, cropland and pasture/rangeland change.** Spatial distribution of **a** forest, **b** cropland and **c** pasture/rangeland extent (stable area) and change (gain and loss) between 1960 and 2019. Area charts on the right show the stacked share of gains, losses and multiple change area (on which both gains and losses have occurred) related to the total area under the respective LUC category along each geographic latitude.

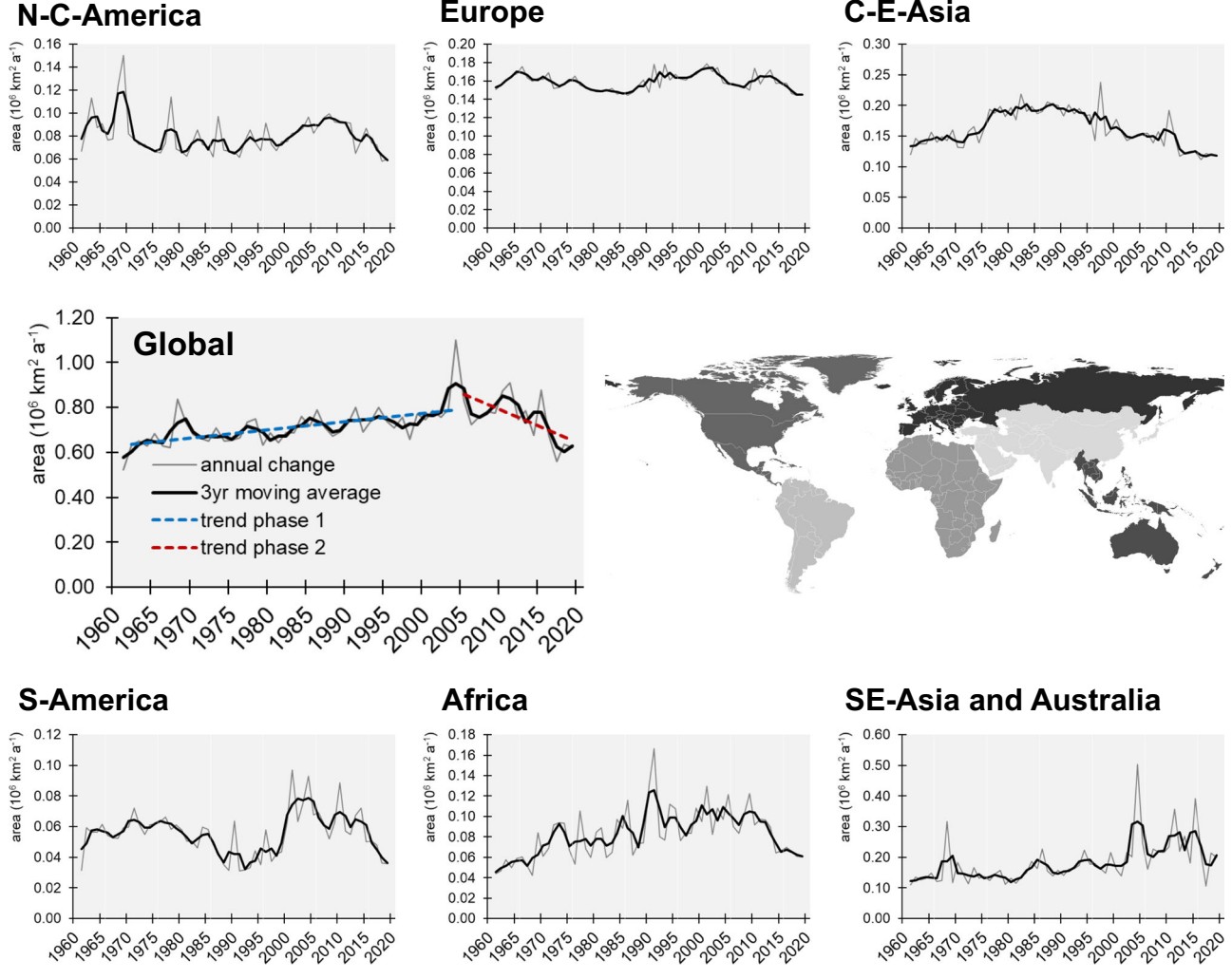

**Fig. 3 Rate of land use change. Annual rate of land use/cover change between 1960 and 2019 for different world regions and the globe.** Global trends are depicted for phases 1: 1960–2004 and 2: 2005–2015. Grey lines show the annual change, black lines show the smoothed annual change based on a 3-year moving average. The map shows the spatial extent of the presented world regions in different shades of grey.

Ethiopia after the 2010/11 drought (see Supplementary Fig. 4). Furthermore, land degradation, caused by both climatic variability and human activities, has often been associated with cropland abandonment, subsequent expansion of agricultural land and deforestation elsewhere, as widely observed in tropical regions[60].

When analysing the temporal dynamics of global land use change per LUC category, we find large annual variability in agricultural land use change. While global forest area shows a rather steady annual net decrease, which accelerated during the 1990s (see Fig. 4a), croplands and pasture/rangeland show large fluctuations over time; about four times higher than observed for forests. This difference likely derives from a combination of the 5-yearly reporting scheme of the FAO/FRA forest data and the quicker response times of agricultural land use change to socio-economic developments. In particular, the rate of agricultural land use change can be affected by political regime shifts (e.g. land abandonment after the collapse of the Soviet Union in 1990)[61], disruptions in globalised supply chains (e.g. the US embargo on soybeans against Russia in 1980)[62,63], nature conservation incentives (e.g. avoided deforestation as in REDD policies)[7], natural hazards and extreme events such as droughts[59,64]. High inter-annual change dynamics in global agricultural land mainly emerged in the 1990s after a long period

of net expansion. This matches the period when major geopolitical shifts (particularly the collapse of the USSR) took place and market-driven food production gained in importance. Whereas pasture/rangelands show a downward trend, which has been attributed to technology advances in the livestock sector[65], global croplands, by contrast, experienced waves of increasing expansion since 2000 (see Fig. 4b, c).

**Comparing the rate of global land use change**. Comparing the HILDA + annual change rates with previous land use reconstructions (see Fig. 4) demonstrates that the area affected by global land use change is nearly four (3.7) times greater than previously thought. A comparison of the rates of land use change between HILDA + and other land use/cover datasets is presented in Fig. 5. Corresponding annual change rates and considered periods are listed in Table 1. Specifically, the mean land use change rate from HILDA + is 2.4 times as high as that of LUH2[14], 4.4 times as high as that of HYDE3.2 and 1.3 times as high as that of SAGE cropland (update from[15]). This deviation is the effect of considering gross changes derived from Earth observation data in HILDA +, which are not or only partially included in the other datasets.

Land cover change rates derived from higher-resolution remote sensing datasets such as Hansen GFC[66], ESA CCI[67] and

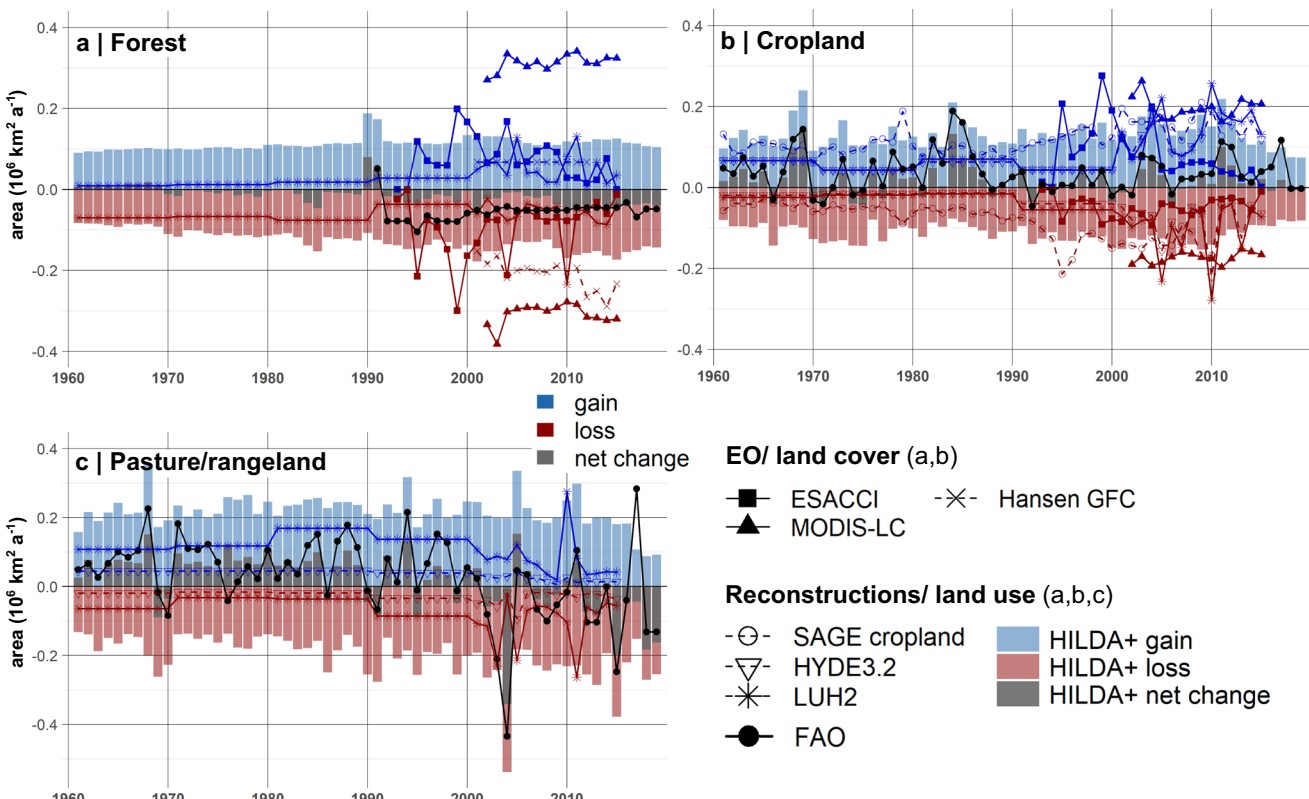

**Fig. 4 Comparison of forest, cropland and pasture/rangeland changes.** Global comparison of annual change of **a** forest, **b** cropland and **c** pasture/rangeland (**c**) (gain, loss and net change area per year) from HILDA +, different Earth observation (EO)-based land cover datasets (ESA CCI[67], MODIS-LC[68], Hansen GFC[66]), land use reconstruction models (SAGE cropland, update from[15]), HYDE3.2[13], LUH2[14]) and FAO land use statistics[41].

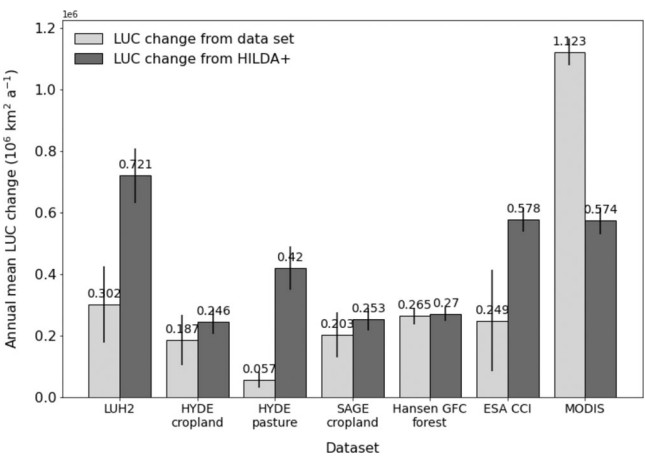

**Fig. 5 Comparison of change rates.** Comparison of mean annual gross land use/cover (LUC) change of different LUC change datasets (light grey bars) with HILDA + (dark grey bars). Error bars represent the standard deviation.

MODIS[68] are on average about the same order of magnitude (1.1 times) as for HILDA +. In particular, the HILDA + annual change rate is on average 1.3 times greater than from remote sensing datasets, with MODIS[68] deviating by +90% ESA CCI[67] deviating by −60% and Hansen GFC[66] deviating by 0% from HILDA + change rates. These differences are most evident for annual forest change rates (see Fig. 4a) and can be explained by different land cover classes on which the original datasets are based, their diverse semantics and delimitations (see Supplementary Tables 1 and 3).

Since HILDA + is built on multiple heterogeneous datasets, errors inherent in single datasets are attenuated during the change allocation procedure. By harmonising multiple information in the change allocation procedure, we build on a confluence of evidence. Thus, HILDA + can be seen as a synthesis product of quality-tested, recognised LUC datasets. To assess the uncertainty of HILDA + maps of global land use change, we analysed the agreement of the used input datasets and the area fraction for each indicated LUC category on an annual basis (see Supplementary Table 4 and Supplementary Figs. 6, 7). Dataset agreement differs per LUC category. Forests and areas with sparse/no vegetation show the highest agreements. On the other hand, dataset deviation is larger in agricultural LUC categories cropland and pasture/rangeland. Especially in heterogeneous landscapes, which hold a mix of managed and unmanaged lands, e.g. savannahs of Sub-Saharan Africa, rangelands in Australia or the grassy steppes of Central Asia, but also in the sparse taiga of eastern Siberia and the transition between Canadian boreal forest and tundra, LUC class coverage is ambiguous (lower area fractions) and, thus, dataset information deviates.

The results of the HILDA + land use change reconstruction show how synergistic information from Earth Observation data, reconstructions and national statistical inventories can be used to identify the spatial patterns and temporal dynamics of global land use change at unprecedented levels of detail. This study shows the benefit of using multiple, data-driven resources, which is needed for comprehensive land change assessment at a global scale. This gives more detailed insights into both the spatial patterns and the temporal dynamics of land use change across the Earth. We identify diverging processes of deforestation and agricultural expansion and demonstrate that the rate and extent of global land use change is responsive to socio-economic developments and disruptions such

**Table 1 Comparison of land use/cover datasets.**

| Dataset | LUC categories included | Compared time period | Annual gross land use change (mean ± standard deviation in 10³ km² a⁻¹) | |
|---|---|---|---|---|
| | | | | HILDA + |
| LUH2[14] | All | 1960–2015 | 302 ± 125 | 721 ± 88 |
| HYDE3.2[13] cropland | Cropland (2) | 1960–2015 | 187 ± 82 | 246 ± 41 |
| HYDE3.2[13] pasture | Pasture/rangelands (3) | 1960–2015 | 57 ± 25 | 420 ± 71 |
| SAGE cropland[15] | Cropland (2) | 1960–2011 | 203 ± 74 | 253 ± 37 |
| Hansen GFC forest[66] | Forest (4) | 2000–2012[a] | 265 ± 27 | 270 ± 21 |
| ESA CCI[67] | All with combined grassland (3 + 5) | 1992–2015 | 249 ± 165 | 578 ± 40 |
| MODIS[68] | All with combined grassland (3 + 5) | 2001–2015 | 1123 ± 44 | 574 ± 43 |

Comparison of annual gross land use/cover (LUC) change (all transitions between included LUC categories or sum of gains and losses for individual LUC categories) of different LUC change datasets with HILDA + for corresponding periods.
[a]Hansen GFC covers forest gain only between 2000 and 2012 (no annual dynamics).

as the global economic crisis 2007–2009. The results suggest that global trade, affecting agriculture, has been one of the main drivers of global land use change over the last six decades.

The HILDA + data have implications for the assessment of climate change, biodiversity loss and food security, especially in estimating carbon budgets, forest management and biomass. Due to its consistent and data-driven change allocation, HILDA + is suited to global time series analysis. Although not free from potential data artefacts, inconsistencies of single datasets are attenuated through the use of multiple data sources. We aim to bridge the gap between long-term FAO-based land use trends, which lack spatial explicitness, and remote sensing-based observational land cover classifications, which lack long-term temporal consistency. Through the synergistic use of observational data and the provision of annual uncertainty measures, HILDA + goes beyond conventional land use reconstructions that often rely on individual datasets, give an incomplete picture of LUC dynamics and lack information about uncertainty. HILDA + provides a consistent time series of global LUC change that provides new possibilities for the analysis of global time series, the identification of possible drivers, impacts and correlations in the context of land use change. Thus, the HILDA + data can contribute to better understanding the environmental impacts of land use change in the past by providing more detailed land change trajectories (e.g. affecting carbon pools) and their temporal classification. It can further improve the assessment of land use strategies in the future in support of policy, e.g. the Paris Climate Targets, the Sustainable Development Goals and the post-2020 agenda of the Convention on Biological Diversity.

## Methods

We reconstructed LUC change dynamics for six LUC categories (urban, cropland, pasture/rangeland, forest, unmanaged grass/shrubland, sparse/no vegetation) based on multiple sources of observational data, from which country-scale change extents and mean fractional area were derived per 1 × 1 km grid cell from 1960 to 2019. We calculated the country- and year-specific areas of change for each land transition between these categories. A base map for the year 2015 served as a starting point for the change allocation procedure, which, at first, runs backward in time (2015–1960) and, subsequently, forward in time (2015–2019). For each time step and country, LUC change was allocated to selected candidate pixels by using ranked gridded class fractions and the data-derived change extents. Each of these iterative procedures yielded a global LUC map, which served as the new base map for the next time step. A visualisation of the HILDA + reconstruction framework, which evolved from the approach of the HILDA over Europe[11], is given in the Supplementary Fig. 8. Methodological steps of the involved change allocation procedure are shown in Supplementary Fig. 9.

**Pre-processing of remote sensing-based LUC data.** The HILDA + reconstruction was derived from multiple, openly available global, continental, regional and national LUC datasets (see Supplementary Table 1).

**Harmonisation of LUC maps.** We defined a common generalised classification scheme for harmonising the remote sensing-based LUC products. The

classification scheme was based on six LUC classes that aim to encompass the major land use changes caused by people and, at the same time, to find a common ground for the input datasets that differ in thematic detail. This classification relates to the FAO land use definitions[17] and the LCCS land cover classification scheme[69] and, thus, combines land cover with land use information. Accordingly, the available LUC maps were reclassified based on their inherent classification schemes (see Supplementary Table 3). The reclassified maps were converted into binary masks for each of the generalised land cover categories. Subsequently, these were reprojected and resampled into the target projection (Eckert IV), the final spatial extent and grid resolution (1 × 1 km) by proportional averaging of the pixel values. Maps of area fractions under each land cover category from Supplementary Table 3 are the result of this processing step.

For those years when no observational datasets were available, remote sensing products with a sufficiently long time series (ESA CCI, MODIS MCD12Q1, GLAD UMD VCF) were back-casted in a stepwise manner, based on a linear extrapolation of the mean trend of the first five observed values in time.

**Probability maps for LUC categories.** For each of the harmonised land cover categories (see Supplementary Table 3) and year of the study period, we derived maps of the average area fractions per grid cell if more than one data source was available. All available datasets were treated as equal. Note that data-inherent uncertainties such as misclassifications, over- and underrepresentation of certain LUC categories in individual datasets are propagated to some degree. However, such inconsistencies are attenuated by relying on multiple datasets instead of a single data source.

Based on the resulting maps of area fractions, we derived probability maps for our final LUC categories (see Supplementary Table 2), which were the basis of the change allocation procedure. The rules for assembling these class probability maps and, on this, converting the generalised land cover maps (see Supplementary Table 3) to our target LUC categories (see Supplementary Table 2) are displayed in Supplementary Table 5.

For separating managed from unmanaged grasslands, we first combined the maps for grassland and shrubland by calculating the mean of their area fractions. We used the resulting maps as probability layers for LUC category 5: Unmanaged grass-/shrubland. For generating the probability layers of LUC category 3: Pasture/rangelands, we used the Gridded Livestock World v3 (GLW, see Supplementary Table 1), which indicates the density of ruminants for the reference year 2010, as an additional indicator of pasture usage. We calculated the mean of the GLW ruminant densities and the area fraction of combined grassland and shrubland categories and used the resulting maps as probability layers for LUC category 3: Pasture/rangelands. Note that, in contrast to grass- and shrubland area fractions, ruminant density information is static (year 2010). Changes in ruminant numbers over time were not considered.

**Base map calibration.** We used the recently released Copernicus LC100 Global Land Cover map for the reference year 2015 to generate a base map for the subsequent reconstruction of LUC change. After reclassifying the map into the generalised land cover categories (see Supplementary Table 3), we reprojected and resampled it into the targeted projection (Eckert IV), spatial extent and grid resolution (1 × 1 km) using majority cell values (mode), resulting in a preliminary land cover map. We calibrated this preliminary base map to FAO national land use statistics for forest, cropland and pasture area[17] using the derived area fractions for each category. The rules applied for the base map calibration procedure are given in Supplementary Table 6.

**Preparing datasets for national LUC change matrices.** The absolute matrices of LUC change, and the land area in each LUC category that changes into another category in a specific country and year, were generated from two different data streams: FAO statistics and remote sensing products. First, we prepared tables of FAO land use area[17] and population statistics[18] per country and year of the study

period. The country extents in the year 2015 were used to ensure a consistent country-specific reconstruction. Thus, land use and population values were completed for countries that have changed in area over the period of 1960–2015 based on trends in the FAO recorded values for the former country before the respective year of change (see Supplementary Table 7). For Europe, LUC values derived from the predecessor HILDA dataset[11] were used to complete the table for periods without FAO data records (e.g. forest before 1990, agricultural areas before 1961). We filled data gaps in the land use table by linear temporal intra- and extrapolation for each country. Secondly, we derived country-specific gross change ratios from transition matrices based on temporally-consistent, long-term, remote sensing-based land cover maps: ESA CCI Land Cover and regional high-resolution datasets for specific regions (CORINE, MoEF Indonesia, AAFC Land Use Canada, NLCD Land Cover, and Australia DLCD). For each country, a mean transition matrix was calculated across all available time steps in the original spatial resolution of the datasets.

**Change calculation**. We derived net changes in the categories 2: Cropland, 3: Pasture/rangelands and 4: Forest from the FAO land use inventories (Arable land and Permanent cropland, Permanent meadows and pastures, Forest), applying the relative changes to the areas from the base map, respectively. We used the base map and the relative population development from FAO (Total population) as a proxy for net urban area change (LUC 1: Urban areas). The remaining land portion (FAO land area minus Urban, Cropland, Pasture/rangelands and Forest area) was divided proportionally into LUC category 5: Unmanaged grass/shrubland and 6: Sparse/no vegetation according to the area ratio of these categories in the base map.

During the change allocation procedure, a new transition matrix including all gross changes between the LUC categories was iteratively built for each time step, each country and each land transition based on the minimum ratio of gross change to class area from the data-derived country-specific mean transition matrix. This ratio represents the average share of land under a specific LUC category that is converted to another category, either a gain or a loss in LUC category.

**Change allocation**. Based on the recalculated country- and year-specific transition matrices, the magnitude of LUC change was distributed over the grid by means of corresponding probability maps for each LUC category. This was carried out in three consecutive steps: First (round 1), change was assigned if the respective LUC categories held the highest area fraction and were greater than 0.1. Second (round 2), if no candidate pixels were found in round 1, change was allocated to grid cells where the area fraction of the respective LUC category was greater than 0.4. Round 3 applied if no candidate pixels were existent after rounds 1 and 2. In the end, no changes were allocated in this step. This procedure was undertaken iteratively for each year (in a back- and forward mode starting from the base year 2015, respectively), for each individual country and for each land transition between two LUC categories. The output of each change allocation step of the annual loop was a new global map of LUC, which served as the base map for the next processing step.

**Change analysis**. The output of the HILDA + change allocation procedure are annual maps of global LUC states (the distribution of LUC categories) and transitions. The transition layers served as the basis for analysing spatial extent, patterns, rates and dynamics of global land use change. Looping through all transition layers, we classified the coded transitions into change and non-change events and counted their occurrence per pixel. The sum of all change occurrences represents the total amount of gross LUC change for the study period. Similarly, LUC category-specific changes were derived by classifying the coded land transitions into gain, loss or stable/non-change events within the respective LUC category. Again, we summed up the occurrences of the different events iteratively through time. Based on the resulting frequencies, we assigned LUC category-specific change on the global grid: gain (single change event), loss (single change event), both gain and loss (multiple change events).

**Uncertainty assessment**. In order to analyse the uncertainty and assess the reliability of the resulting HILDA + dataset, we derived annual layers of uncertainty information based on the available input LUC datasets. The number of available datasets, the maximum deviation in class area fraction and the mean class area fraction from all available datasets per year were used to generate per-pixel quality information. Based on the multi-year mean of dataset agreement (maximum deviation) and class coverage (mean class area fraction), global quality flags were derived and mapped across the globe (see Supplementary Table 4, Supplementary Figs. 5 and 6).

## Data availability
Source data from remote sensing, land use reconstructions and statistics used in the HILDA + model are listed and described in Supplementary Table 1. The dataset generated and analysed during the current study, the HILDA + Global Land Use Change dataset (vGLOB-1.0), is available in the PANGAEA repository, (https://doi.org/10.1594/PANGAEA.921846) as Open Data[70]. Visualisation of the HILDA + Global Land Use Change data is provided in form of an interactive map viewer (https://landchangestories.org/hildaplus-mapviewer/). Further background information and stories accompanying the HILDA + project are published on a blog, www.landchangestories.org.

## Code availability
The reconstruction modelling and analyses were performed using Python 3.7. Computer codes for the development and analyses are available upon request to the corresponding author. Future releases will be communicated through the HILDA + map viewer (https://landchangestories.org/hildaplus-mapviewer/).

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

## Acknowledgements

This research has been supported by the European Commission, Horizon 2020 Framework Programme (VERIFY, grant no. 776810). K.W. was supported by the Open Science Fellows Programme of Wikimedia Deutschland e.V. (2019–2020). We acknowledge Christian Werner for helpful advices on developing the interactive map viewer to visualise our data.

## Author contributions

K.W. and R.F. created the study concept and designed the research. K.W. collected, processed and analysed the data. K.W. and R.F. formed the HILDA + framework and K.W. developed the model. K.W. prepared the manuscript, with support from R.F. K.W. created all figures and maps. All authors interpreted and discussed the results. All authors were involved in critical revision of the manuscript, and commented on the paper. M.R. guided the progress of the study and made linguistic edits. M.H. supervised the study.

## Funding

## Competing interests

The authors declare no competing interests.
