## [Peer Review File · Nature Communications]

REVIEWER COMMENTS

Reviewer #1 (Remarks to the Author):

Authors assessed annual changes in land use/cover from 1960 to 2015 based on multiple remote sensing data (variable spatial resolution) and statistical data. They found that global land use change has affected over a third of the global land area in just six decades (1960-2015) and, thus, is around four times greater in extent than previously assumed from long-term land change assessments. This study advances previous studies in terms of spatial resolution and number of land cover/use categories considered. It is well written and provides detailed information on the methods used. I invite authors to consider the two concerns below in revising the paper.

(1) Compared to existing global, long term land use reconstructions, which authors indicate rely on few observations (lines 43-45), this analysis rely on a significant number of global/regional LULC datasets. On one hand, I think it is advantageous to consider more observations. But on the other hand, more observations could introduce errors and confusion in the harmonisation and subsequent change analysis. The datasets presented in supplementary Table 1 are not only diverse in spatial resolution, thematic detail, etc., but also diverse in terms of the mapping accuracies due to differences in underlying data (including reference/groundtruth data) and methods/algorithms used. I'm, for example, aware of huge underestimation of cropland areas in some of the datasets (e.g. GlobeLand30 in West Africa – Forkuor et al. 2017), while others may perform slightly better. I did not quite see how authors considered these diverse accuracies or representations in the harmonisation of the maps, and how these errors may have propagated to the subsequent change analysis. Authors seem to have treated them as equally accurate, which I'm not sure is the case. And the diversity of these accuracies and representations can easily give the wrong impression of changes, especially for cropland in heterogeneous landscapes in Africa (line 185-186). That aside, I noticed that classes representing vegetation/cropland mosaics (e.g. class 14 for MODIS) were classified as cropland per Supplementary Table 2. While I appreciate why authors will make this choice, I'm wondering what the impact of such choices will have on the final analysis, and whether extra analysis was not required to provide error estimates for changes observed. Considering the detailed and somewhat complex procedure in determining the annual changes (plus the FAO statistics), I suspect some level of error propagation, which should be accounted for somehow. Unless I missed or misunderstood this, I highly recommend that authors think through this and provide some error estimates on the the observed changes or at least an admission of possible errors in the analysis.

(2) I think that authors dwelt too much on global trade and economic crises as possible reasons for observed LULC changes. It seems to me that climate change and associated impacts (e.g. floods, droughts, land degradation) were somewhat downplayed, especially for developing countries with lower adaptive capacities. For example, in West Africa, devastating floods, which impact cropland, have been a constant feature between 2006 and 2015 (period of deceleration). In fact, 2007 and 2009 happen to be two of the worse years, which could also be the reason for observed LULC changes (see lines 146-147). That aside, land degradation is a major reason for land abandonment and expansion of cropland into forest areas in Africa. Unfortunately, the paper seems not to mention land degradation at all. Unless I completely misunderstood the argumentation related to trade and economic crises, I strongly recommend that authors do a bit more investigation to determine other possible causes for observed changes, especially in agricultural land/areas.

References

Forkuor, G., Conrad, C., Thiel, M., Zoungrana, J.B., Tondoh, J.E., 2017. Multiscale Remote Sensing to Map the Spatial Distribution and Extent of Cropland in the Sudanian Savanna of West Africa. *Remote Sensing* 9 (8). doi:10.3390/rs9080839.

Reviewer #2 (Remarks to the Author):

I scrutinized with great interest this manuscript which, evaluates a temporal and spectral

intercomparison land use change analysis using HILDA+. I acknowledge the data analysis however; major corrections need to be addressed.

General comments:

- 1- Please avoid long and confusing sentences. Use concise and simple scientific language to communicate your findings.
- 2- Please Improve the quality of figure 1 and 2.
- 3- Please revise the manuscript for grammar flaws.
- 4- What is/are the conclusion (s) drawn from this study? Is HILDA+ suitable for the global temporal change analysis? Please provide a concise conclusion from these findings.

Specific comments

To talk food security and climate change, LULCC assessment methodology must go beyond quantity assessment (what is driving the change? and what can be done to alleviate the change?). There is a need to analyze and integrate the drivers of the changes; evaluate the intensity and the speed of the changes and provide point-based scale assessment for each time-slice across different agroecological zones rather than global scale. Therefore, some gaps from this manuscript need to be addressed:

- 1- Analyze and provide a temporal analysis of the dominant proximate drivers (e.g. cropland management, deforestation rate, afforestation measures, literacy, poverty....) of land use change in addition with the population data.
- 2- Provide a LULCC vulnerability maps of each dominant drivers using a matrix of change analysis and specific weighted coefficient.
- 3- Provide an overlay intensity analysis maps of each vulnerability category to discuss the temporal changes in the LULC using HILDA+.

RESPONSE TO REFEREES

The reviewers' comments are shown in grey.

Authors' answers are in bold.

REVIEWER #1

(Remarks to the Author)

Authors assessed annual changes in land use/cover from 1960 to 2015 based on multiple remote sensing data (variable spatial resolution) and statistical data. They found that global land use change has affected over a third of the global land area in just six decades (1960-2015) and, thus, is around four times greater in extent than previously assumed from long-term land change assessments. This study advances previous studies in terms of spatial resolution and number of land cover/use categories considered. It is well written and provides detailed information on the methods used. I invite authors to consider the two concerns below in revising the paper.

Dear Reviewer #1,

Thank you very much for giving your attention to our manuscript and for your critical review. Based on your helpful comments and constructive suggestions, we have revised our manuscript. In doing so, we think that we have been able to improve the quality of the study.

General information:

Since submitting our manuscript, we have worked on an updated version on HiLDA+ - now extended in time and covering the period of 1960-2019. The new HiLDA+ dataset comes with the following adjustments:

- Integration of newly available datasets: Copernicus Land Cover 100m 2016-2019, CORINE Land Cover 2012/ 2018 (ESA CCI Land Cover 2016-2018 forest classes for version with forest sub-division(dynamics))
- Updated FAO land use database, annual national land use trends based on FAOSTAT land use, updated version 2020-09-10
<http://www.fao.org/faostat/en/#data/RL/metadata>
- Changes in ESA CCI legend harmonisation: Sparse vegetation classes (150-153; see Supplementary Table 2) are assigned to HiLDA+ class 6: Sparse/no vegetation (important for change allocation procedure)
- More stringent change allocation rules for reducing “mismatch” between observational and reported data streams

We revised all plots, figures and numbers according to the analysis of this new HiLDA+ version. This has led to slight changes in the global extent and overall spatiotemporal patterns of land use change. We see modifications of some regional and national trends in land use change (e.g. annual change rates of Brazil and China). However, the overall storyline and findings of the study remain the same.

In the following, we will give you a point-to-point response to your comments provided:

(1) Compared to existing global, long term land use reconstructions, which authors indicate rely on few observations (lines 43-45), this analysis rely on a significant number of global/regional LULC datasets. On one hand, I think it is advantageous to consider more observations. But on the other hand, more observations could introduce errors and confusion in the harmonisation and subsequent change analysis. The datasets presented in supplementary Table 1 are not only diverse in spatial resolution, thematic detail, etc., but also diverse in terms of the mapping accuracies due to differences in underlying data (including reference/groundtruth data) and methods/algorithms used. I'm, for example, aware of huge underestimation of cropland areas in some of the datasets (e.g. GlobeLand30 in West Africa – Forkuor et al. 2017), while others may perform slightly better. I did not quite see how authors considered these diverse accuracies or representations in the harmonisation of the maps, and how these errors may have propagated to the subsequent change analysis. Authors seem to have treated them as equally accurate, which I'm not sure is the case. And the diversity of these accuracies and representations can easily give the wrong impression of changes, especially for cropland in heterogeneous landscapes in Africa (line 185-186).

Data-inherent accuracy and uncertainty are indeed important points to address. Not all datasets provide accuracy layers and if so, they are fragmented. In some cases, datasets come with overall accuracies (~73.6% for MODIS MCD12Q1 v006 ¹), accuracies for single reference years (~71-71 % for ESA CCI Land Cover in 2015 ², 67.7 % and 70.7 % for GlobCover in 2005 and 2009 ³, 68.6 % for GLC 2000 ⁴, or 76.5 % for GLCNMO in 2003). In others, they provide additional per pixel quality flags, which consist of broad categories (“good” and “other quality” in MODIS MCD12Q1 v006 ⁵). Due to this heterogeneity in accuracy information, we treated the datasets indeed as equally important for the respective time step in the change allocation procedure of the presented HiLDA+ version.

We are aware of mapping inconsistencies of some of the used input data such as the mentioned underestimation of cropland areas from Globeland30 in West Africa. In the methodology of HILDA+, we generated area fractions as an information of “common agreement” based on averaging the fractions from multiple data sources per time step (if available) for each land use/cover (LULC) category. Therefore, possible inconsistencies of a single dataset were not transferred directly to the change allocation procedure. They are attenuated due to harmonising and synthesising the available information – a “confluence of evidence” effect. That means that no dataset is taken as the only reference. Ranking of suitable pixels based on those mean LULC fractions was done iteratively for each transition. The data harmonisation and ranking procedure weakens the effect of single, possibly inconsistent or erroneous data sources. However, our harmonised HiLDA+ dataset is still constrained by the accuracy of the input datasets and, thus, large inconsistencies inherent in an input dataset, albeit weakened (due to the averaging), might still come into effect.

To take into account potential inconsistencies of datasets (compared to others) we derived additional uncertainty information. This includes layers of the annual deviation of class area fractions from each available dataset. Here, the range (maximum deviation) between the datasets is used to describe the agreement of the datasets concerning the respective LUC category from HILDA+.

That aside, I noticed that classes representing vegetation/cropland mosaics (e.g. class 14 for MODIS) were classified as cropland per Supplementary Table 2. While I appreciate why authors will make this choice, I'm wondering what the impact of such choices will have on the final analysis, and whether extra analysis was not required to provide error estimates for changes observed.

Several land cover classification data used in this study provide mixed or mosaics of land use/cover classes such as MODIS or ESA CCI. Based on TSENDBAZAR ET AL. 2017 ⁶, we decided to include all mosaic classes in cropland, where crops cover at least 40 % of the total land cover. We added this more detailed description of the cropland category and its definition in Table 2 and 3 in the Supplementary Information.

Testing the impact of this and any other choice related to land use/cover categories on the final analysis is complex, since it would involve a very large number of model runs and goes beyond the scope of the study.

Considering the detailed and somewhat complex procedure in determining the annual changes (plus the FAO statistics), I suspect some level of error propagation, which should be accounted for somehow. Unless I missed or misunderstood this, I highly recommend that authors think through this and provide some error estimates on the the observed changes or at least an admission of possible errors in the analysis.

Regarding the error estimates, providing accurate error propagation and unravelling the contribution of each individual step, such as the choice of land use/cover definitions, dataset-inherent uncertainties or thresholds used in the allocation procedure, is difficult due to the structure of the change allocation model. Despite, to address this very valid point, we calculated the per-pixel agreement between the used input datasets to provide an estimate of uncertainty based on the observational data sources (see section Methods: “Uncertainty assessment” lines 419-427). For each land use/cover category from HiLDA+ between 1960 and 2019, the levels of agreement between the used input datasets are provided. These include the average area fraction, the deviation and the number of available datasets for each time step and land use/cover category, respectively. Uncertainty patterns and distribution is briefly discussed in section “Results and discussion” (see lines 243-256).

(2) I think that authors dwelt too much on global trade and economic crises as possible reasons for observed LULC changes. It seems to me that climate change and associated impacts (e.g. floods, droughts, land degradation) were somewhat downplayed, especially for developing countries with lower adaptive capacities. For example, in West Africa, devastating floods, which impact cropland, have been a constant feature between 2006 and 2015 (period of deceleration). In fact, 2007 and 2009 happen to be two of the worse years, which could also be the reason for observed LULC changes (see lines 146-147). That aside, land degradation is a major reason for land abandonment and expansion of cropland into forest areas in Africa. Unfortunately, the paper seems not to mention land degradation at all. Unless I completely misunderstood the argumentation related to trade and economic crises, I strongly recommend that authors do a bit more investigation to determine other possible causes for observed changes, especially in agricultural land/areas.

References

Forkuor, G., Conrad, C., Thiel, M., Zoungrana, J.B., Tondoh, J.E., 2017. Multiscale Remote Sensing to Map the Spatial Distribution and Extent of Cropland in the Sudanian Savanna of West Africa. *Remote Sensing* 9 (8). doi:10.3390/rs9080839.

Due to your suggestions, we expanded the discussion about underlying drivers of land use change and its temporal dynamics. We found that, especially for West Africa and Ethiopia, climate change-related extreme events such as droughts have affected land use and its rate observed from around 2005. Furthermore, land degradation was added as a potential driving factor of land change dynamics. Our presented study aims at providing a first synthesised database on global land use change and its full annual dynamics. With opening the HILDA+ dataset to the public, we intent to encourage further in-detail analyses on drivers, interrelations and impacts of land use change on a global scale. This first paper focuses on spatiotemporal patterns, extent and timing of land use change itself. We are currently working on a follow-up study, which revolves around the processes and drivers of land use change and goes into more detail about the influencing factors of land use change and their interrelations.

REVIEWER #2

(Remarks to the Author)

I scrutinized with great interest this manuscript which, evaluates a temporal and spectral intercomparison land use change analysis using HiLDA+. I acknowledge the data analysis however; major corrections need to be addressed.

Dear Reviewer #2,

Thank you for studying our manuscript and for your critical and valuable remarks. Based on your comments and constructive review, we have revised our manuscript to the best of our understanding. We believe that this has allowed us to improve the quality of the study.

General information:

Since submitting our manuscript, we have worked on an updated version on HiLDA+ - now extended in time and covering the period of 1960-2019. The new HiLDA+ dataset comes with the following adjustments:

- Integration of newly available datasets: Copernicus Land Cover 100m 2016-2019, CORINE Land Cover 2012/ 2018 (ESA CCI Land Cover 2016-2018 forest classes for version with forest sub-division(dynamics))
- Updated FAO land use database, annual national land use trends based on FAOSTAT land use, updated version 2020-09-10
<http://www.fao.org/faostat/en/#data/RL/metadata>
- Changes in ESA CCI legend harmonisation: Sparse vegetation classes (150-153; see Supplementary Table 2) are assigned to HiLDA+ class 6: Sparse/no vegetation (important for change allocation procedure)
- More stringent change allocation rules for reducing “mismatch” between observational and reported data streams

We revised all plots, figures and numbers according to the analysis of this new HiLDA+ version. This has led to slight changes in the global extent and overall spatiotemporal patterns of land use change. We see modifications of some regional and national trends in land use change (e.g. annual change rates of Brazil and China). However, the overall storyline and findings of the study remain the same.

In the following, we will provide a point-to-point response to each of your remarks:

General comments:

1- Please avoid long and confusing sentences. Use concise and simple scientific language to communicate your findings.

We went through the text again and shortened some longer and potentially confusing sentences.

2- Please Improve the quality of figure 1 and 2.

The quality of Figure 1 and 2 was improved by increasing the resolution of the maps.

3- Please revise the manuscript for grammar flaws.

We revised the manuscript once more for grammar mistakes

4- What is/are the conclusion (s) drawn from this study? Is HILDA+ suitable for the global temporal change analysis? Please provide a concise conclusion from these findings.

We revised and updated the concluding paragraph (from line 258). With HILDA+, our intention was to build a suitable dataset for global time series analysis with a consistent change assessment and data-driven methodological structure. We aim to bridge the gap between long-term FAO-based land use trends, which lack spatial explicitness, and remote sensing-based observational land cover classifications, which lack long-term temporal consistency. In our revised manuscript and dataset version, we provide additional uncertainty information in form of the agreement and deviation of the used input datasets. HILDA provides new possibilities for follow-up studies about e.g. the identification of possible drivers, impacts and correlations in the context of land use change. We added a more concise explanation on the potential of the results from HILDA+ to the concluding paragraph (from line 272).

Specific comments

To talk food security and climate change, LULCC assessment methodology must go beyond quantity assessment (what is driving the change? and what can be done to alleviate the change?). There is a need to analyze and integrate the drivers of the changes; evaluate the intensity and the speed of the changes and provide point-based scale assessment for each time-slice across different agroecological zones rather than global scale. Therefore, some gaps from this manuscript need to be addressed:

We agree on that. The question on the potential drivers of land use change is highly relevant and has to be discussed besides the quantity assessment. However, it is beyond the scope of our presented paper, which is explicitly about the extent, timing and patterns of global land use change. A detailed assessment of the drivers of land use change is in progress and we plan to present it in a follow-up study. Nevertheless, we extended the discussion about the drivers of land use change and its dynamics. Aside from globalised trade, we mention climate change and associated impacts such as droughts and/or land degradation (from line number 180).

According to your suggestions, we analysed the spatiotemporal patterns of land use change in form of the annual rate of change for different agro-ecological zones (see Supplementary Figure 6) and included it to our analysis.

1- Analyze and provide a temporal analysis of the dominant proximate drivers (e.g. cropland management, deforestation rate, afforestation measures, literacy, poverty....) of land use change in addition with the population data.

A detailed analysis of dominant drivers of change is complex. Thus, we decided to split the assessment of the drivers and the detailed description of the land use change processes from the quantitative analysis of global land use change, its extent, spatial and temporal dynamics. We are currently preparing a follow-up study specifically on land use change

processes, patterns and major drivers. Additionally, with publishing the HILDA+ dataset as Open Data to the public, we explicitly encourage further usage and exploitation of the data for a more detailed assessment of the direct and indirect drivers of land use change. We added some statements to clarify on the objectives and limitations of HILDA+ to the discussion (see lines 272-283).

2- Provide a LULCC vulnerability maps of each dominant drivers using a matrix of change analysis and specific weighted coefficient.

(See comment below)

3- Provide an overlay intensity analysis maps of each vulnerability category to discuss the temporal changes in the LULC using HILDA+.

Thank you for suggesting the approach of vulnerability maps to study the importance of the drivers. We find that this subject opens up a new research question and wide field of methods to address it. In our view, the topic of the drivers of change goes beyond the simple question on the amount, timing and patterns of global land use change we presented here. We would like to address this issue separately in a follow-up study and, in doing so, provide a more detailed assessment of the drivers of change.

REFERENCES

1. Sulla-Menashe, D., Gray, J. M., Abercrombie, S. P. & Friedl, M. A. Hierarchical mapping of annual global land cover 2001 to present: The MODIS Collection 6 Land Cover product. *Remote Sensing of Environment* **222**, 183–194 (2019).
2. ESA. *Land Cover CCI Product User Guide Version 2*.
maps.elie.ucl.ac.be/CCI/viewer/download/ESACCI-LC-Ph2-PUGv2_2.0.pdf (2017).
3. Bontemps, S. *et al.* GLOBCOVER 2009-Products description and validation report. *URL: http://due.esrin.esa.int/files/GLOBCOVER2009_Validation_Report_2.2.pdf* **2**, (2011).
4. Mayaux, P. *et al.* Validation of the global land cover 2000 map. *IEEE Transactions on Geoscience and Remote Sensing* **44**, 1728–1739 (2006).
5. Friedl, M. A. & Sulla-Menashe, D. MCD12Q1 MODIS/Terra+Aqua Land Cover Type Yearly L3 Global 500m SIN Grid V006. (2019).
6. Tsendbazar, N.-E., de Bruin, S. & Herold, M. Integrating global land cover datasets for deriving user-specific maps. *International Journal of Digital Earth* **10**, 219–237 (2017).

REVIEWERS' COMMENTS

Reviewer #1 (Remarks to the Author):

Dear Authors,

Thank you for paying attention to my comments and revising the manuscript accordingly. I'm pleased with the revision and congratulate you in advance for your contribution to knowledge

Reviewer #2 (Remarks to the Author):

Thank you for including some of our comments and suggestions in improving this manuscript.